# Renal Papillary Necrosis (RPN) in an African Population: Disease Patterns, Relevant Pathways, and Management

**DOI:** 10.3390/biomedicines11010093

**Published:** 2022-12-29

**Authors:** Guy Roger Gaudji, Meshack Bida, Marius Conradie, Botle Precious Damane, Megan Jean Bester

**Affiliations:** 1Department of Urology, Steve Biko Academic Hospital, Faculty of Health Sciences, University of Pretoria, Pretoria 0007, South Africa; 2Department of Anatomical Pathology, National Health Laboratory Service (NHLS), Faculty of Health Sciences, University of Pretoria, Pretoria 0084, South Africa; 3Urology Practice, Netcare Waterfall City Hospital, Cnr Magwa Avenue and Mac Mac Road, Johannesburg 1682, South Africa; 4Department of Surgery, Steve Biko Academic Hospital, Faculty of Health Sciences, University of Pretoria, Pretoria 0007, South Africa; 5Department of Anatomy, Faculty of Health Sciences, University of Pretoria, Pretoria 0007, South Africa

**Keywords:** renal papillary necrosis, coagulative necrosis, genetic, diabetes, HIV, schistosomiasis free radicals, oxidative stress

## Abstract

Renal papillary necrosis (RPN) is characterized by coagulative necrosis of the renal medullary pyramids and papillae. Multiple conditions and toxins are associated with RPN. Several RPN risk factors, or POSTCARDS, have been identified, with most patients presenting with RPN having at least two contributing risk factors. Currently, there is no specific test to diagnose and confirm RPN; however, several imaging tools can be used to diagnose the condition. RPN is currently underdiagnosed in African populations, often with fatal outcomes. In African clinical settings, there is a lack of consensus on how to define and describe RPN in terms of kidney anatomy, pathology, endourology, epidemiology, the identification of African-specific risk factors, the contribution of oxidative stress, and lastly an algorithm for managing the condition. Several risk factors are unique to African populations including population-specific genetic factors, iatrogenic factors, viral infections, antimicrobial therapy, schistosomiasis, substance abuse, and hypertension (GIVASSH). Oxidative stress is central to both GIVASSH and POSTCARDS-associated risk factors. In this review, we present information specific to African populations that can be used to establish an updated consensual definition and practical grading system for radiologists, urologists, nephrologists, nuclear physicians, and pathologists in African clinical settings.

## 1. Introduction

Renal papillary necrosis (RPN) was first described in 1877 by Dr. Nikolaus Friedreich [1], who first noticed the condition in patients with prostatic hypertrophy and secondary hydronephrosis. Six decades later, Froboese [2] and Günther [3] found that RPN was associated with diabetes mellitus and later described the condition in patients with urinary tract obstruction without diabetes mellitus. In 1952, Mandel [4] observed that 95% of urinary tract infection cases were associated with RPN and specifically with analgesic abuse, which accounts for approximately 85% of all cases [5]. Currently, the description of RPN by clinicians and radiologists does not correlate with prognostic outcome, especially in the African context. Most vulnerable population at risk of chronic kidney diseases (CKD), acute kidney injuries (AKI), acute tubular necrosis (ATN) and RPN in African setting appear to be mothers and children exposed to poor socioeconomical background [6]. Despite the very limited published data available on the pattern and pathways of RPN in African population, the hallmark of the disease depends on geographical location, environmental hazards and the quality of basic health care system [7,8,9]. The most morbid risk factors are unknown and underdiagnosed genetic diseases such as congenital obstructive nephropathy (posterior urethral valves) [10], and sickle cell nephropathy [11]. The primary defect of sickle cell disease results from the replacement of glutamate for valine at the sixth amino acid of the beta globin chain. Subsequently the mutation lead to hemoglobin S which accumulate due to tissue hypoxia, dehydration, and oxidative stress [12]. Other risk factors including diabetic nephropathy, infectious diseases (TB, HIV, COVID-19), analgesic nephropathy, iatrogenic, severe immunosuppression, poor compliance on HAART, and substances abuse will be discussed in more detail in the text.

Although acute renal failure and associated pathophysiology has been described in detail; we still have limited understanding of the causes of acute tubular necrosis [13]. Currently, the relationship between renal cell tubular cellular changes in acute kidney injury and clinical presentations remains vague. Diagnostic imaging results can, however, be analyzed in combination with appropriate biological markers, including cytokines and their receptors such as tumor necrosis factor (TNF) receptor one [14,15], chemokines, and growth factors. Several biomarkers, including kidney injury molecule-1 (Kim-1), renal papilla antigen (RPA) 1 and 2, have been correlated with antibiotic use and metal nephrotoxicity [16]. Preclinical evidence suggests that RPA-1 may be a useful non-invasive diagnostic marker for early stage nephrotoxicity and RPN [17]. Further studies have also found Kim-1 to be more reliable for diagnosing RPN than serum creatinine, blood urea nitrogen, and neutrophil gelatinase-associated lipocalin (NGAL) [18]. Similarly, NGAL can be used to differentiate prerenal failure and acute tubular necrosis.

Currently used investigative methods, including abdominal computerized tomography (CT) scan, urea, and creatinine glomerular filtration rate (GFR) can only describe late changes, disease sequelae and the global impact of multiple insults, and do not reflect the contribution of emerging disease, drugs, and other contributing factors. Several additional factors common to African populations have been identified and reviewed. Many of these factors may contribute to the development of RPN in South African clinical settings. We discuss the role of oxidative stress which is central to existing risk factors for RPN and the additional factors that are specific to African populations. In addition, we propose an algorithm for disease management.

## 2. Renal Pyramid, Renal Papilla, and Vasa Recta: A Unique Morphological Entity

The function of the kidneys is to regulate body fluids and electrolytes, remove waste products and drugs, control red blood cell production, and release hormones that regulate blood pressure and active vitamin D for healthy bones [19]. The renal papillae are located at the top of medullary pyramids and the papillary ducts are just below the apex. Below the renal papilla is the renal calyces, which includes the renal fornix. The renal papilla transports urine from the kidneys via the cortical section to the lumen of the calyx, from which the urine then drains from the pelvis to the bladder via the ureter. Within a typical kidney, there are 7–9 renal papillae, although this number can range from 4 to 18 per kidney. Each papilla is adjacent to a cup-shaped minor calyx that drains the urine into the renal pelvis [19,20], Figure 1A.

Blood is directly supplied to the renal papilla via the vasa recta which is the terminal artery of the renal artery, which branches into the lobar, interlobular, arcuate, and ultimately the interlobular arteries. The interlobular arteries supply the glomeruli via the afferent arterioles. The vasa recta branches into three terminal branches at the bottom of the medullary pyramid. The branches become narrower once the apex and papilla are reached. This limits the blood supply within the papillary tip, hence the risk for ischemia and, consequently, RPN is increased [21], Figure 1B. The loop of Henle is referred to as counter current multiplier and vasa recta as a counter current exchange system in concentrating and diluting urine, hence the increased risk of oxidative stress damage [22]. In the sagittal view, the kidney is divided into the outer cortex and the inner medulla. The former entails distal and proximal convoluted tubules, glomeruli, medulla rays, interlobular arteries, and veins. Each pyramid base is located next to the cortex and its apex forms the renal papilla that opens up into the minor calyx. A columnar epithelium covers the tip of the papilla merging into the transitional epithelium. The renal pedicle and branches; veins, arteries, and interlobular vessels are located within the renal sinuses. The interlobar vessels divide into the arcuate artery and vein which gives rise to smaller interlobular arteries and interlobular veins that supply the cortex branching into the afferent glomerular arteries and subsequent capillaries that branch to the glomeruli [23]. Ischemic biological necrobiosis of the papilla in the medulla of the kidneys causes cell death of the renal medullary pyramids and papillae. The hallmark of the disease is coagulative necrosis of the renal medullary pyramids and papillae secondary to multiple conditions (POSTCARDS) and toxins [24]. Most patients who present with papillary necrosis have at least two contributing risk factors, and necrosis can involve one or up to 18 papillae [25]. The focal type of RPN involves only the tip of the papilla whilst the diffuse type of RPN involves the whole papilla and some sections of the medulla. The pathological features are characterized by the degree of ischemia and can be divided into the (a) medullary form: described as intact fornices, discrete grain sized necrotic areas, and later defects in the papilla, sinus tracts extruding from irregular medullary cavities, and (b) papillary form: calyceal fornices damaged, faceless, and necrotic [21].

## 3. Diagnostic Shortcomings


**A.** **Clinical Features** 


Symptomatic clinical presentation of RPN includes the following: fever and chills, flank, and/or abdominal pain. Hematuria is almost always present, with acute ureteral obstruction from sloughed papillae that manifests as flank pain. Colic due to hydronephrosis or pyonephrosis can also occur. Besides fever and chills, sepsis may also occur. Complications include infection, obstruction as well as the presence of calculi [26]. Nevertheless, non-symptomatic RPN which is associated with the unknown risk factors often has the most lethal outcome. For example, patients who have had kidney transplant and later developed kidney stones present with different clinical features than those of non-transplant patients. The presence of renal colicky pain is not expected and rarely mimics acute rejection. Therefore, the need for renal transplantation biomarkers that correlate with cold ischemic time, serum creatinine, predictors of acute tubular necrosis, and long-term graft function are crucial [27,28].


**B.** **Imaging** 


There is no specific test currently used to diagnose and confirm RPN. Several imaging tools are used to diagnose RPN including ultrasound and CT scans (Figure 2 and Figure 3). Ultrasound scan features of RPN are a non-shadowing soft tissue mass in the ureter, a clot, sludge, or fungal ball, and the late phase showing necrosed papillae in the medullary cavities. Cavitation zones in the papillae are a number of either circles or triangular shaped cystic spaces in the medulla surrounding renal sinus echo. Calcified echogenic sloughed papillae and cast shadows obstruct the collecting system with subsequent hydronephrosis [29]. More over, with CT scan, findings are small kidneys with ring shadows in the medulla, contrast-filled clefts in the renal parenchyma, and renal pelvic filling defects [30]. In addition, endoscopic tools such as ureteroscopy, percutaneous approach, and biopsy further confirm the diagnosis [31,32].


**C.** **Cytopathology** 


The histological diagnosis of RPN is sometimes difficult in cases of subtle ischemic lesions as opposed to definite changes in the toxic form of the disease. In addition, the location of tubular damage differs between ischemic versus toxic form associated with the loop of Henle [13]. Chronic infections including human immunodeficiency virus (HIV) can result in a persistent inflammatory response that would eventually lead to chronic inflammation (Figure 4B), a condition known to exacerbate kidney disease. Oxidative stress is one of the key factors of CKD as it is involved in promoting chronic inflammation known to promote tissue necrosis [33] (Figure 4C). In RPN, necrosis may occur throughout the medulla but would mostly be at the papillary tip which is the most susceptible due to the narrowing of the vessels as they reach the papilla. Consequently, this is the site where the first damage is observed. For example, urinary obstruction may lead to pigmentation, focal ischemia, necrosis, and hemorrhage of papilla [34]. As necrosis is induced by ischemia, it typically has a coagulative appearance with the presence of pinocytes, spherocytes, and ghost cells as observed in Figure 4D. Microcalcifications may also be present [35].

## 4. Potential Diagnostic Paradigm for RPN

A patient suspected with RPN based on clinical findings (e.g., hematuria) and standard diagnostic imaging (ultrasound and CT scan) may need further special investigations such as endourological procedures including, cystoscopy, pyelogram, or selective renal unit decompression as a resuscitation tool. Then, additional investigative tools can be used, e.g., open tip urine sample collection for cytology, oxidative stress markers, cytokine levels, and renal pelvis barbotage with normal saline to enhance the yield of cytology results and lastly defining ureteroscopic RPN features after biopsy [32] (Figure 5). Cystoscopic access allows the selective collection of urine and slough papilla for cytology, pathology, biochemistry, and immunology thus providing additional diagnostic criteria to radiology and nuclear medicine [36,37].

## 5. RPN Epidemiology in The World and Sub-Saharan Africa

RPN particularly caused by analgesic drug abuse, is more prevalent in Australia and England compared to America accounting for 15–20% of patients who require renal transplant [38,39,40]. A high incidence of RPN was observed in African patients with homozygous sickle cell disease (SCD) [41]. Contradictory to this study, Madu et al. [9], found a low prevalence of symptomatic RPN in a similar group of South Eastern Nigerian patients. Generally, the prevalence of asymptomatic or symptomatic RPN is found to occur at a similar rate in SCD [42,43]. The prevalence of SCD is significantly higher in African countries such as Uganda with a rate difference observed according to districts [44]. In South Africa, the rate of SCD is increasing mainly due to people migrating from other African regions with the Democratic Republic of Congo representing the highest prevalence at 62.1% [45]. Data on the association between RPN and other genetic disorders such as renal tubular acidosis is lacking. In addition, the contribution of other infections such as HIV, schistosomiasis, and TB to the development of RPN is unknown, especially considering the burden of disease in this region.

## 6. RPN Etiologies and Risk Factors


*A. Etiology of Renal Papillary Necrosis*


Known causes of RPN include diabetes nephropathy (diffuse nodular mesangial sclerosis), analgesic nephropathy (interstitial fibrosis, tubular atrophy, capillary sclerosis), and sickle cells (sickling in blood vessels). The mnemonic for RPN is “POSTCARDS” and the common causes are listed Table 1 [46]. Two or more risk factors increases the risk of RPN.

However, POSTCARDS do not account for all the factors associated with the development of RPN especially in the African context. Evaluation of the scientific literature, revealed several additional factors that could be associated with the development of RPN in South African populations. These include genetic disorders, iatrogenic factors, antibiotic use, schistosomiasis, substance abuse, and hypertension (mnemonic GIVASSH) (Figure 6). A better understanding of the contribution of these factors, early intervention, and more effective management may prevent the development of RPN and associated complications. The identified factors are listed in Table 2 and each will be discussed in greater detail with a focus on the prevalence and impact on the South African population.

G-Genetic

Medullary sponge kidney, cystic kidney diseases and gout lead to RPN in the African population but remain underdiagnosed. The former condition also known as ectatic papillary duct leads to end-stage renal kidney disease if undiagnosed and treated late. It is proposed that the architecture of the vasa recta contributes to the concentration of urine along with the loop of Henle and the collecting duct. In South Africa, the 3′ region of PKD1 mutation is the molecular genetic basis of autosomal dominant polycystic kidney disease (ADPKD) associated with polycystin-1 and polycystin-2 dysfunction implicated in tubulogenesis changes reported in South Africa, Tunisia, Senegal, Sudan, and Congo [48]. In the Afrikaner population, a founder mutation, apM627 K substitution at the PKHD1 locus is associated with Autosomal Recessive Polycystic Kidney Disease (ARPKD) [49]. Although sickle cell anemia is associated with a high prevalence of glomerulonephropathy, other genetic diseases such as renal tubular acidosis Type I causing RPN remain under diagnosed in African populations [50,51,52].

The apolipoprotein L1 (APOL1) renal risk variant is present in African but absent in Asian and European populations. Evidence for the APOL1 association with CKD in Sub-Saharan Africa is compelling. According to the candidate gene approach, two gene defects have been associated with kidney chronic disease which are the myosin heavy polypeptide 9 (MYH9) and APOL1 polymorphisms in populations of African ancestry [53,54]. There is also a strong correlation between APOL1 risk genotype, G1/G2, and cytokines such as tumor necrosis factor receptor (TNFR) 1 or 2 and kidney injury molecule-1 (KIM-1) [55]. Identification of population specific mutations and renal risk variants, highlights the role of geneticists and genetic testing in the diagnosis of renal disease and associated risk factors.

I-Iatrogenic

Although not confirmed, urological intervention might have an impact on the progression of CKD. Nevertheless, level 4 evidence from expert endourologists have identified the nephrotoxicity of contrast agents causing contrast induced nephropathy. Other factors such as intra-abdominal pressure, radiation, guide wire use, stents, basket, and laser lithotripsy may also contribute to the development of RPN [56,57,58,59]. Examples include a forgotten ureteral stent due to lack of patient education or poor follow-up after ureteral stenting. The management of neglected stents needs sophisticated complex endourological procedures for which experts are usually not available in Africa [60]. Indeed, some urological training programs in Sub-Saharan Africa have limited endoscopic training capacity. In addition, some centers still consider open surgery with its high morbidity on the kidney [61]. The narrowed therapeutic window of calcineurin inhibitors (CNI) immunosuppression after renal transplantation is associated with long-term nephrotoxicity, hypertension, and metabolic disorders including the following features on histology: epithelial necrosis, thrombotic microangiopathy, interstitial fibrosis, tubular atrophy, and premature graft failure [62]. Furthermore, in black African patients higher required doses of tacrolimus further increases the risk for RPN [63].

V-Viral

In a 2019, an estimated 7.9 million South Africans were living with HIV. A total of 13.5% of the population of which most are on highly active antiretroviral therapy (HAART) [64]. The nephrological manifestation of HIV can present as a collapsing variant of focal segmental glomerulosclerosis. Additional nephropathies such as ATN and thrombotic microangiopathies have been described [65]. A large number of kidney disorders are related or indirectly related to HIV infection. This includes antiretroviral (ARV)-induced acute kidney injury, proximal tubular dysfunction, and crystalluria among others [66]. Moreover, the nephrotoxicity of ARVs is difficult to assess in African clinical settings. Previous studies on the accuracy of the Cockcroft Gault equation, which assesses the drug dosing in HIV patients yielded conflicting results and did not use ideal biomarkers such as inulin or radioisotope clearance in South Africa and Ghana [67]. Nephrotoxicity is associated with the development of RPN, thus novel approaches can potentially limit the HAART nephrotoxicity [66,68].

Most HIV-infected people present with opportunistic infections and are commonly co-infected with TB. The morphologic features of kidney TB include pyelonephritis, papillitis, papillary necrosis, calyceal blunting, and multifocal cystic daisy flower pattern [69]. The pathophysiology of HIV-associated RPN follows a particular pattern. This includes AKI leading to obstructive AKI by cast then acute tubular injury (ATI) resulting in ATN precisely focal diffuse tubular epithelial cell coagulative type necrosis. On histological findings, there is evidence of tubular cell cytoplasm swelling and appearance of ghost-like tubular forms [70]. Furthermore, an unusual case of acute papillary necrosis was recently reported in a COVID-19 patient with necrosis attributed to the virus. Inflammatory features similar to those depicted in Figure 2B and the presence of fluffy tissue in almost all of the renal calyces similar to Figure 2C were observed. This case somehow validated the importance of incorporating screening for viral infections in patients suspected of RPN [71]. Therefore, the impact of new emerging infections such as COVID-19 on renal function and RPN should be considered. Other infections such as Hepatitis C can also lead to glomerulonephritis secondary to cryoglobulinemia, proteinuria, vasculitis, and fibrinoid necrosis [72].

Most nephrotoxicity related to HAART is caused by metabolic acidosis and Fanconi syndrome. Many children in SA are on HAART and the long-term effects on the kidneys are not yet known. However, Soares et al. [73] assessed the connection between HAART treatment and kidney disease in HIV-infected children. The study found that 33 (52.3%) out of 63 patients had acute kidney injury (AKI) but that children who were on HAART had a higher estimated GFR, indicating that HAART protected against AKI. Their findings suggest that suppressing viral infection may prevent kidney damage [73].

A-Antimicrobials

Besides the effects of ARVs, and the well-known nephrotoxicity of aminoglycoside, there is little published data on the nephrotoxicity of new generation antibiotics such as quinolone and carbapenems [74]. Side effects of fluoroquinolone include granulomatous interstitial nephritis which is characterized by infiltration of histocytes and T cells resulting in granuloma formation. In some instances, cases of acute tubular necrosis secondary to this antibiotic have been reported [75]. The World Health Organization discourages the use of remdesivir in COVID-19 patients precisely because of the hepatorenal toxicity observed in some patients [76].

S-Schistosomiasis

Several parasitic infections are associated with kidney disorders [77]. Urinary schistosomiasis is a disease caused by the parasitic worm, *Schistosoma haematobium*. In South Africa, more than 4 million people are estimated to have schistosomiasis [78]. Patients present with hematuria, hepatosplenic, and specific urogenital disorders including glomerulopathy, focal segmental glomerulosclerosis, chronic interstitial nephritis, and fibrosis with a typical early triad, loss of urine concentration, sodium wasting, and tubular acidosis. Schistosomiasis and glomerular disease pathophysiology includes immune-mediated glomerulonephritis (antibodies), oxidative stress-mediated tubular injury, urinary tract obstruction, hepatic fibrosis (immunoglobulin A), an autoimmune mechanism as well as a genetic predisposition to schistosomal infection [79,80]. The long-term morbidity of urinary schistosomiasis in children is diverse and often devastating and includes acute or chronic anemia, hypertension, end-stage renal disease and fatal metabolic disorder, and death [81]. Schistosomiasis causes fibrosis of the ureter and subsequent obstructive uropathy, reflux nephropathy, nephrolithiasis, and pressure necrosis [82,83] which predisposes the patient to RPN. Urinary schistosomiasis manifests as a typical obstructive uropathy in the upper and lower urinary tract and superimposed damage by acute pyelonephritis, reflux nephropathy, and undiagnosed congenital pyeloureteric junction (PUJ) stenosis resulting in RPN pressure mediated injury [82]. In agreement with these findings, Hong-Hong et al. [84] found a strong link between obstruction and RPN confirmed by ultrasound, CT scan, and endoscopy.

S-Substance Abuse

Smoking induces glomerulopathy and tubular dysfunction and also reflects the direct damage of nicotine and other toxic molecules present in cigarette smoke [85]. An average of 17.6% of the adult South African smoke [86]. Van Heerden et al. [87] described substance use in South African populations and found that 38.7% consume alcohol, 30% smoke tobacco, 8.4% use cannabis, 2% drugs, and 19.3% psychoactive drugs. Renal damage secondary to substance abuse includes atherosclerotic and ischemic damage, intestinal inflammation, calcification, intestinal fibrosis, and tubular atrophy in relation to the specific insult [88] with specific nephrotoxicity in synthetic cannabinoid users [89]. The cocktail drug, a novel psychoactive substance, nyaope consists of 10–70% third-grade heroin added to antiretroviral drugs, mixed with cannabis [90]. Also depending on the region and raw material the composition could include tetrahydrocannabinol, diamorphine, caffeine, dextromethorphan, phenacetin, efavirenz, or nevirapine [91]. In addition, in these drug users there is often a superimposed HAART nephrotoxicity [92].

H-Hypertension

The prevalence of hypertension ranges from 42–54% in South Africa. Of those who are diagnosed and are on treatment, only about 39% were reported to have controlled blood pressure [93]. Benign and malignant hypertension cause nephropathy; ischemic tubular atrophy, glomerulosclerosis, and nephrosclerosis [94]. People with lifestyle patterns that include inadequate physical activity and consumption of unhealthy food such as sweetened beverages, salty or fast food are more likely to develop hypertension than people who are physically active and opt for healthier meals [95]. Of 556 black South African study participants, 71% had hypertension significantly associated with diabetes, westernized diet (includes fast food), and obesity. Socioeconomic status has been identified as a contributing factor [96]. This is concerning as hypertension is a known significant risk factor for the development of CKD and end-stage renal disease [97,98], hence possible association with RPN.

## 7. Contributing Pathways to RPN

### 7.1. Acute Tubular Necrosis Cumulative Effect Theory

The pathophysiology of ATN in African populations occurs according to the following pathway cascade. First, anatomical variation of the nephron can cause filtration failure. The dyssynergia between glomerular filtration and proximal loop reabsorptive function is associated with increased sodium chloride in the macula densa, and a subsequent decrease in renal blood flow and the glomerular filtration rate. Subsequent tubular epithelial damage causes the back leak of filtered tubular fluid into the vascular space. Consequently, apoptotic changes are observed in the proximal and distal tubular cells. Congested leukocytes in the outer medulla affect local renal blood flow. Additional processes contributing to ATN are endothelial and epithelial injury, intratubular obstruction, impaired local microvascular blood flow, and inflammatory or immunological responses [99,100,101].

### 7.2. Pathogenesis of RPN in the Course of Analgesic Nephropathy

Analgesic nephropathy (AN) is chronic tubulointerstitial nephritis caused by long-term use of analgesics such as over the counter pain medication and nonsteroidal anti-inflammatory drugs (NSAIDs) [102]. Other compounds such as caffeine or codeine which serve as additives in some medications have also been implicated [103]. This also raises concerns regarding the high caffeine content in energy drinks meant to assist with high endurance and keeping people awake. Nonetheless, indication of analgesics as a major etiological factor of RPN has long been described (1950s) with 80–90% of RPN cases attributed to AN [24]. Previous studies report an increased number of RPN cases in people who abuse aspirin, with some studies describing paracetamol AN, caused by the main metabolite phenacetin. The mechanisms of phenacetin-induced RPN include the tendency to cluster and concentrate within the renal papilla causing injury to the tip of the papillae [104]. The pathophysiology of AN starts with the thickening or hardening of the vasa recta capillaries with dispersed tubular necrosis (as in the case of ATN) followed by papillary necrosis, secondary cortical necrosis, interstitial inflammation, and fibrosis [105].

Analgesic abuse has always been one of the major risk factors of kidney disease in South Africa [106]. Ten South African patients with AN were found to be on different medications with aspirin, paracetamol, and codeine being the most common. All the patients had retrograde pyelogram evidence of RPN with two patients passing slough papilla as confirmed by a pathologist [36]. Several comorbidities including AN were found to be the cause of CKD in 21 cases confirmed by means of tissue biopsy [107]. Among 328 study subjects, 40% where at high risk of developing CKD due to analgesic abuse [108]. In another study, 22 participants with kidney dysfunction were frequent users of NSAIDS. Six percent of these participants had confirmed kidney disease. The risk of kidney disease was higher in subjects who were also on herbal medicine [109]. Another study reported that most patients first present with an unknown past medical history and with late stage disease. Lastly, these patients have poor outcomes after frequent use of nephrotoxic herbal medication prescribed by traditional healers. The end result of AN in these population is synergistic toxicity, cumulative dose-dependent or idiosyncratic dose-independent toxicity [110]. There is poor awareness amongst general practitioners of the superimposed renal tubular damage caused by using NSAIDs in combination with anti-TB treatment and HAART.

The mechanism of NSAIDs-induced AN includes alteration of renal hemodynamics via inhibition of the cyclooxygenase (COX) enzymes leading to ischemia. The main function of COX enzymes is to catalyze prostaglandin synthesis from the arachidonic acid pathway. This anti-prostaglandin effect lower glomerular filtration rate and disrupt regulation of renal blood flow/pressure and sodium excretion which have a significant influence on renal function. The Cox enzymes stimulate the production of prostaglandins needed for maintaining a homeostatic balance between water and sodium, vascular tone, and stimulate secretion of enzyme renin, responsible for the regulation of renal blood pressure [111,112,113]. It is recorded that about 20% of patients who are on non-selective NSAIDs have multiple risk factors including high NSAID dosage, heart disease, diabetes etc., that can lead to renal dysfunction [111].

## 8. Reactive Oxygen Species and RPN

Inflammation is triggered by oxidative stress with a large body of evidence indicating that renal tubular epithelial cell hypoxia is the final common pathway to end-stage renal failure with reactive oxygen species (ROS)/reactive nitrogen species (RNS) predominantly being responsible for kidney injury. The rationale for measuring ROS/RNS is the lack of obvious clinical findings in the early phase of kidney disease. In addition, renal parameters such as urea and creatinine levels are usually elevated only during the late stages of renal disorder. Several factors including diet, nephrotoxins, uremic toxins, and other comorbidities contribute to renal dysfunction. Subsequently, ROS/RNS accumulation will occur long before nitrogenous substances begin to rise in the bloodstream [22,114]. Failure to regulate an influx of inflammatory markers triggered by high levels of ROS could result in persistent inflammation and renal fibrosis, Figure 7. Oxidative stress could result in apoptosis or necrosis whereas obstruction causes only apoptosis in distal tubular cells and the interstitial space [115]. These are some of the key factors in the development of renal damage and consequent progression to CKD. Oxidative stress is reported as a risk factor for mortality in CKD. Not only does it promote kidney disease progression but also leads to comorbidities such as diabetes mellitus and cardiovascular diseases [33,116].

The impact of ROS/RNS in the kidney is specific to the various parts of the nephron. First, in the medullary thick ascending limb, nicotinamide adenine dinucleotide phosphate (NADPH) is the main supplier of superoxide anions including hydrogen peroxide, hydroxyl radical, and nitric oxide. The reabsorption of sodium ions (Na^+^) is significantly affected by the dysfunction of the apical Na^+^/K^+^ (potassium ion)/Cl^−^ (chloride ion) (NKCC) co-transporter. Second, ROS triggers the absorption of Na^+^ in the distal tubule and collecting duct via two more mechanisms. The epithelial Na^+^ channel and NADPH oxidase maintains ROS reabsorption in this area. Changes in the oxidative stress pathway disturbs the absorption of Na^+^ leading to kidney disease [117].

Diabetic nephropathy is an example where oxidative stress and associated pathways have been identified as therapeutic targets for the effective management of RPN. Glucose is not effectively utilized in these patients and ends up being excreted in urine. Consequently, this imbalanced glucose hyper filtration is corrected via two mechanisms: an increase in basal Na^+^/K^+^ ATPase and Na^+^-dependent glucose transporter 2 (SGLT2). However, mitochondrial tubular dysfunction secondary to severe hypoxia produces ROS (superoxide) that induces papillary necrosis via apoptosis, autophagy, pyroptosis, and release of advanced oxidation protein products [118].

The potential therapeutic target for ROS kidney damage depends on the various sources from endo-and exogenous factors [119], Figure 8. Therapeutic drugs such as paracetamol, bleomycin, and doxorubicin can also induce ROS production in various diseases [120]. Cellular antioxidant systems that modulate the negative effects of ROS include antioxidant molecules such as glutathione and the upregulation of the antioxidant enzymes, superoxidase dismutase, cytosolic catalase, and glutathione peroxidase. In clinical trials and clinical practice, promising outcomes have been reported with the use of anti-oxidant drugs such as vitamin C and E [121], allopurinol [122], melatonin [121], transforming growth factor β inhibitors [123], nuclear factor erythroid 2-related factor 2 activators (NRF2) [124], and co-enzyme Q_10_ (CoQ_10_) [125].

The endoplasmic reticulum (ER) provides a suitable environment needed for oxidative protein folding, assembly, and formation of disulfide bonds. Proper protein folding is facilitated by the production of ROS species such as hydrogen peroxide within the ER. However, the ER can be overwhelmed resulting in improper protein folding which in turn causes ER stress, Figure 8 [126]. Cells undergoing ER stress develop unfolded protein response (UPR) with an intension to correct ER stress. However, if ER stress remains unresolved, UPR could lead to oxidative stress, inflammation, and consequently apoptosis. The ER stress is associated with several diseases including diabetes- and kidney-associated disease such as ischemia-reperfusion injury [127]. Ischemia-reperfusion injury of podocytes takes place through ER stress (PERK/eIF2α phosphorylation) signaling pathway [128]. Studies show that NSAIDs, particularly indomethacin, induces ER stress in murine podocytes [129].

Patient compliance toward preventative screening measures is a challenge that could be solved by identifying non-invasive ROS biomarkers that patients can measure in their own homes. Easily obtainable samples such as saliva and urine can be used to access oxidative markers such as ferric reducing activity, levels of advanced oxidation protein products, and uric acid. However, there are disadvantages and challenges regarding the sensitivity and specificity of these methods [114]. Nevertheless, measuring oxidative stress levels may assist with diagnosing and managing RPN.

The development of ROS-related renal fibrosis involves several mechanisms. This includes the Kelch-like epichlorohydrin-associated protein 1, NRF2 signaling pathway which contributes to oxidative injury-induced nephropathies [130]. ROS play an important role in ischemia that leads to RPN and associated renal fibrosis. Ischemia-reperfusion injury (IRI) causes tissue damage and experimental studies show that IRI-induced kidney fibrosis is facilitated by high levels of ROS [131] (Figure 9). Transforming growth factor-beta in association with ROS is significantly associated with renal fibrosis [132] and can therefore be considered as a potential indicator of renal fibrosis that occurs as a consequence of ROS-mediated activation of fibroblasts. Several studies have discussed the organ-specific aspect of ROS/RNS mediated fibrosis due to fibroblast activation and subsequent non-specific deposition of extracellular matrix [133]. Renal fibrosis can be considered as the final consequence of improper repair mechanisms as well as maladaptive repair resulting in interstitial fibrosis, tubular atrophy, and capillary rarefaction. Mitochondria produces about 90% of ROS and subsequently mitochondrial damage leads to ROS overproduction which causes kidney damage and disease progression [134,135,136] (Figure 9). In addition, an association between the intact basement membrane and toxic tubular degeneration or renal necrosis has been described in canines and these findings are similar to that seen in renal disorder post-exposure to antibiotics, NSAIDs, and myoglobinuria [137,138]. Fibrous tissue stains such as Trichrome, Sirius red, or Ladewig can be applied to evaluate the extent of fibrosis in the glomerulus or the tubular interstitium [139,140,141]. Standard histopathological methods involve general staining with hematoxylin and eosin (Figure 4B–D) and may not always identify early fibrotic changes in the basement membrane. Currently, imaging tools for RPN cannot identify the threat and associated complications of acute or CKDs. Consequently, identifying additional factors contributing to RPN in South African populations as well as more sensitive biochemical and histological, and biochemical methods will contribute to early diagnosis, early implementation of therapy, and the prevention of RPN complications.

## 9. RPN Management; Medical and Surgical Therapy

The major insult causing RPN is ischemia, therefore fluid challenge and respiratory support treating hypoxia together with optimization of the past medical and surgical disorders are mandatory. In addition, intravenous broad-spectrum antibiotic cover, sugar control, and urine alkalization are required. In the meantime, stop all nephrotoxic drugs and diets which may improve renal function [26]. Urgent and adequate urine drainage via ureteral stenting or percutaneous nephrostomy are valid surgical options. Subsequently, a systemic antibiotic should be administrated while waiting for Gram stain and blood culture results. Thereafter, treatment can be adjusted according to sensitivity [142].

## 10. Summary, Conclusions and Future Prospects

Some schools of thought consider RPN as a misnomer. Indeed, the false positive diagnosis varies between 10 and 30%. The ultimate aim is to prevent late-onset disease and the need for kidney transplant where the three-year graft survival range in sickle cell disease is 46–59% in different series [143,144]. Although the traditional causes have been identified (Table 1), additional factors should be considered in the context of Africa (Table 2). A further confounding factor is that patients often have more than one risk factor. These risk factors are usually undetected or undiagnosed due to a lack of population-based screening programs in urology, cardiovascular disease, and transmissible disease at various levels of the health system. Genetic disorders missed during childhood include examples of anecdotal cases of renal tubular acidosis (RTA), Fraley syndrome, and adult polycystic kidney disease associated with RPN in addition to the lethal association with RPN related iatrogenic injury and viral disease such as HIV including nephrotoxic therapy. Clinical presentation of RPN worsens in the presence of acute urinary obstruction, supported by the cumulative effect theory in biology, particularly in acute kidney injury and multiple organ failure.

Early diagnosis of RPN remains a challenge. RPN is highly underdiagnosed in African setting. Most of the time RPN is diagnosed incidentally. GIVASSH is not intended to replace POSTCARDS, it is the additional potential risk factors observed in African population. The components thereof stem mostly from clinical observations. Other clinicians and scientists in Africa or similar settings can further explore this concept and prove or oppose it based on their own experience and clinical findings. This will add knowledge and challenge clinicians in low-middle-income countries to broaden their minds when treating patients with multiple factors related to kidney disease. For instance, HIV is associated with opportunistic infections. These co-infections are the major contributing risk factors to RPN. Examples include urogenital TB, bacterial emphysematous pyelonephritis, and fungal infection. The risk is increased in individuals with low CD4 count and high viral load related to poor compliance. In African settings, patients tend to consult traditional healers before presenting to healthcare professionals. These healers prescribe non-approved nephrotoxic herbal medication. Although there are a few Caucasian traditional healers practicing traditional medicine, this is mostly unique to the black African population and still is practiced in great numbers. Patients from rural areas or poverty-stricken areas in South Africa are the most non-compliant. Apart from believing that HIV is witchcraft, most patients reside far from the hospital making it a challenge for them to come for hospital follow-ups due to lack of money or transport. Substance abuse is also a challenge. Fear of the disclosure of HIV status makes it difficult for these patients to take their medication at home as they might have to explain to family members what the medication is for [145,146].

Consensus reports acknowledge that adequate structural data in kidney disease may improve the global outcome grading system hence tissue morphology analysis may be of value. Less invasive methods that measure changes in oxidative status and stress, associated inflammation biomarkers, pathways in the glomerulus and endothelium may help early detection. These biomarkers of tubular necrosis have a potential prognostic role in renal failure classification and evaluation [147]. Indeed, they would complement the gaps in the scoring system regarding sepsis [148], trauma [149], fibrosis and postoperative surgical outcome [150]. These findings can assist to assess the perioperative pitfalls for renal access [percutaneous nephrolithotomy (PCNL) and renal sparing surgery]. Innovative and emerging radiological tools, including ultrasound and Doppler could enable better diagnosis of RPN. Additional methods could be considered for early diagnosis and management of RPN, Figure 10.

## Figures and Tables

**Figure 1 biomedicines-11-00093-f001:**
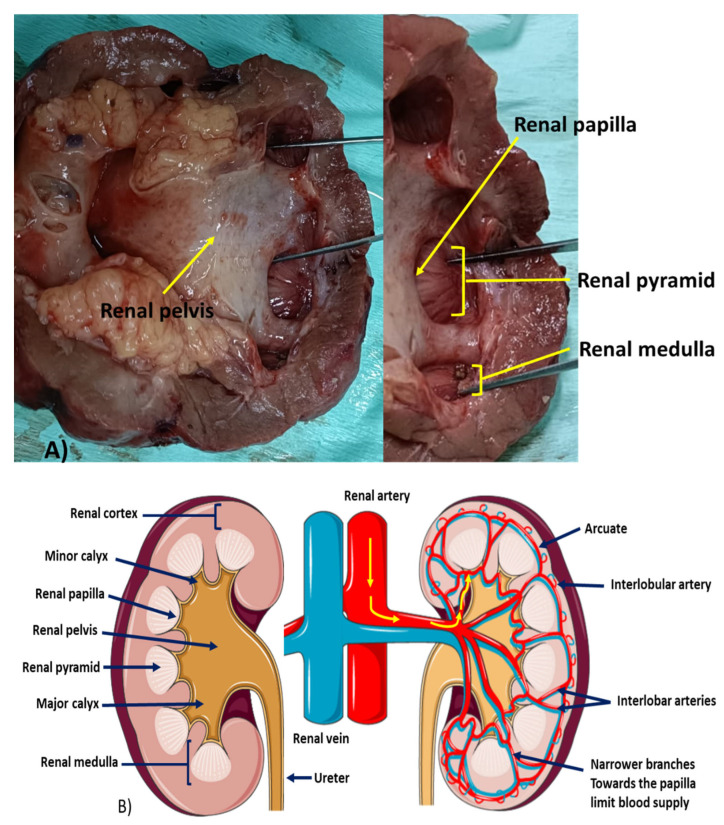
An illustration of renal papilla and fornix. (**A**) Depicts the position and structure of the renal papilla which lies within the renal medulla; and (**B**) the blood supply to the renal papilla showing that as the arteries narrow toward the papilla, the blood flow reduces predisposing the papilla to ischemia which could lead to necrosis.

**Figure 2 biomedicines-11-00093-f002:**
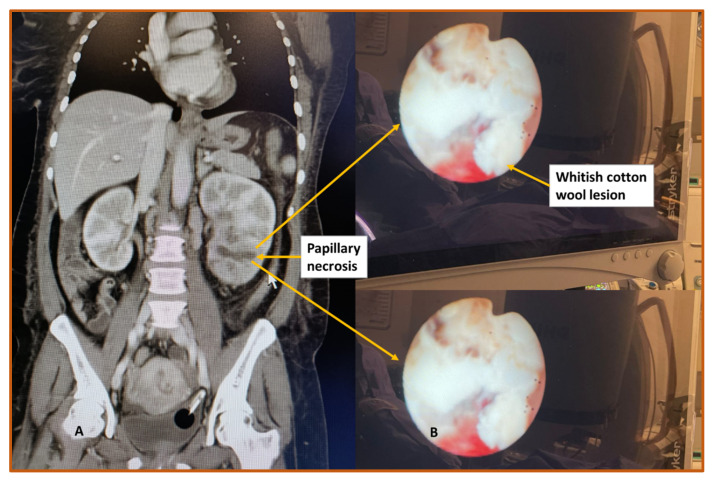
Features of renal papillary necrosis (RPN) in a 50-year-old female. (**A**) The computed tomography (CT) scan indicates spongiform lesion in a lower pole and posterior midpole regions. There are multiple small locules consistent with tubulo-interstitial nephritis or RPN. The patient was diagnosed with human immunodeficiency virus (HIV), pulmonary tuberculosis (TB) and immune reconstitution inflammatory response syndrome (IRIS); (**B**) the whitish cotton wool type lesion seen with flexible ureteroscopy in the lower pole of the left kidney.

**Figure 3 biomedicines-11-00093-f003:**
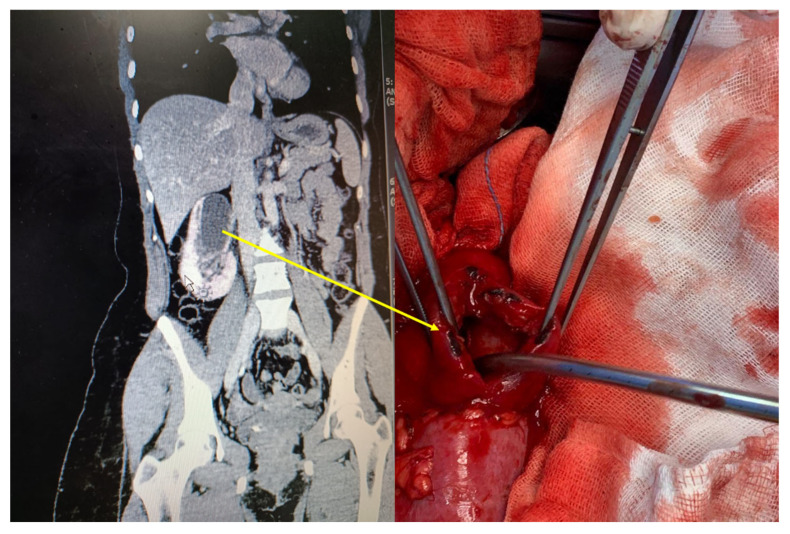
Features of RPN in Fraley syndrome. This is a histologically confirmed case of RPN after partial nephrectomy. The CT scan denotes the upper pole obstruction with hydrocalycosis as depicted by the arrow. The per-operative findings image post excision are also shown.

**Figure 4 biomedicines-11-00093-f004:**
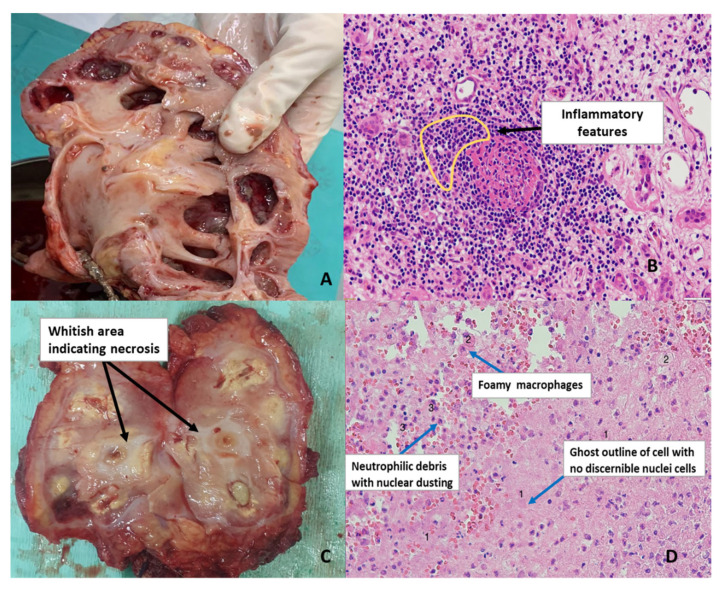
Gross anatomy and histological images showing. (**A**) Kidney with RPN that appears largely fibrotic with marked chronic interstitial nephritis; (**B**) inflammatory features are shown in the drawn region as a feature of neutrophil infiltration. In places, aggregates of mononuclear lymphocytes form mature lymphoid follicles. Plasma cells are also present; (**C**) an image of a kidney with RPN. The indicated white regions are due to lack of blood supply which cause tissue necrosis; (**D**) an image depicting coagulative necrosis. Ghost cells are denoted by the number 1, 2 depicts foamy macrophages and 3 is neutrophil debris.

**Figure 5 biomedicines-11-00093-f005:**
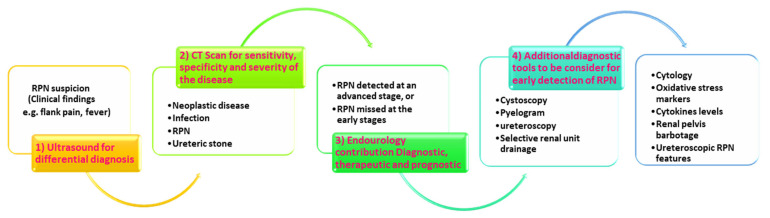
Emerging diagnostic paradigm shift for RPN diagnosis. Several approaches could be considered when investigating RPN. (1) Ultrasound is used for differential diagnosis which could detect advanced RPN or other diseases such as ureteric stone; (2) CT scan is more sensitive and mostly specific in detecting kidney diseases but early stages of RPN can still be missed at this point; (3) endourological approach could be more determinant during an emergency (hematuria) or elective procedure; (4) additionally, other diagnostic assays could be investigated as reliable methods that could assist in detecting and diagnosing the RPN early and allow sufficient time to prevent further damage to the kidney.

**Figure 6 biomedicines-11-00093-f006:**
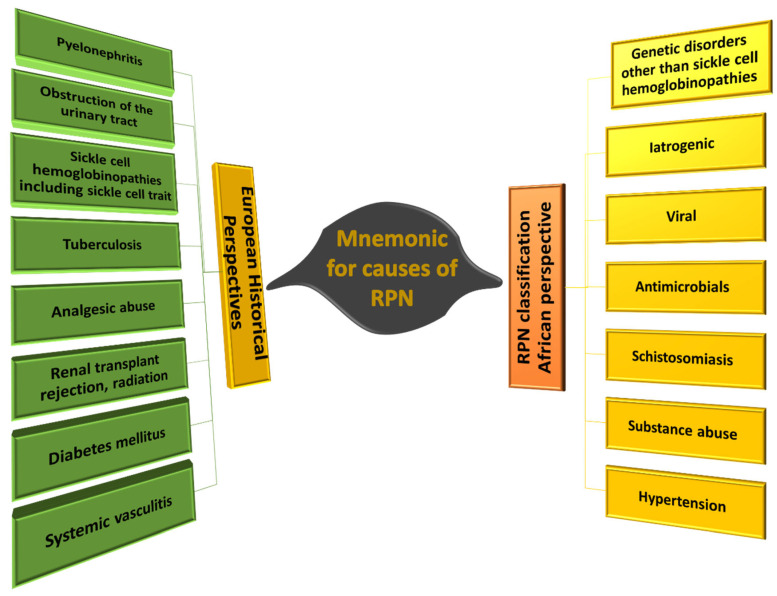
Additional suggested mnemonic RPN risk factors. Histologically, RPN is characterized by several features with the main initiator being coagulative necrosis of the renal papilla and the medullary pyramids. These features are still a valuable diagnostic representation of RPN as listed in POSTCARDS. However, several African patients present with additional features that should be considered to allow clinicians to tackle the disease more effectively.

**Figure 7 biomedicines-11-00093-f007:**
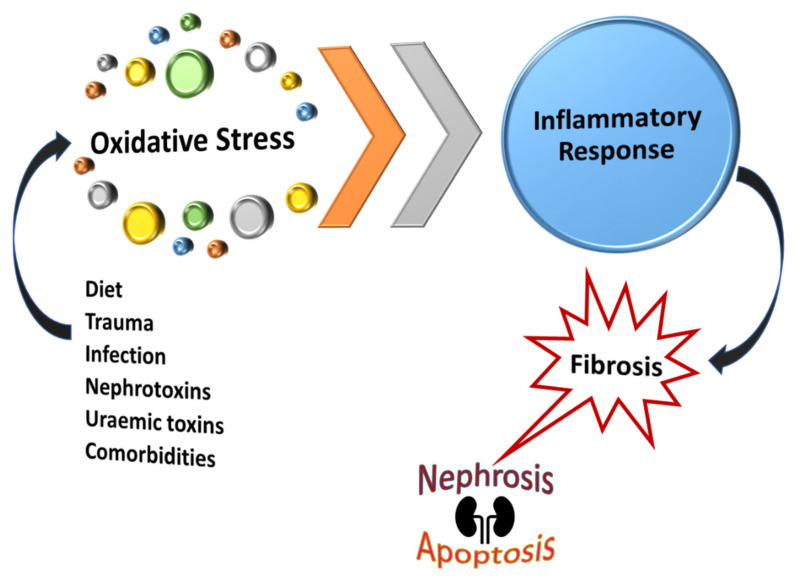
Oxidative stress in acute and chronic kidney disorders is associated with the development of RPN and fibrosis. A high concentration of oxidative stress products is associated with several toxic factors, subsequently triggering an inflammatory response, and leading to kidney damage.

**Figure 8 biomedicines-11-00093-f008:**
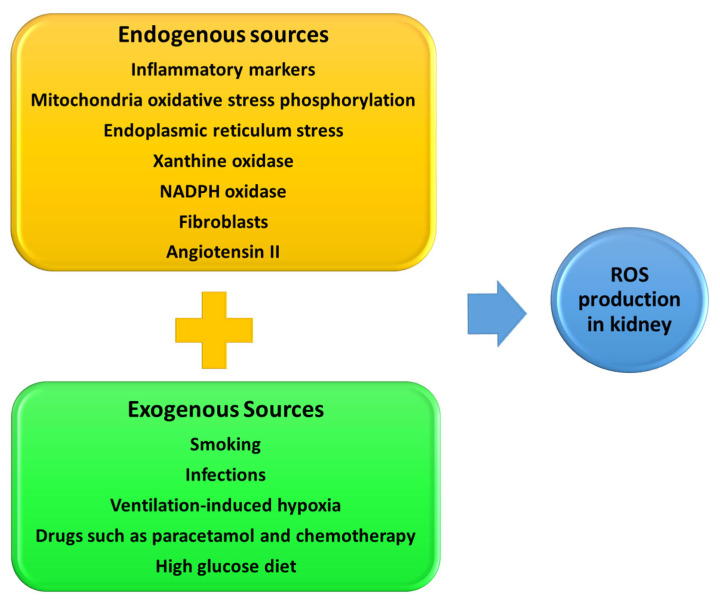
Major sources of reactive oxygen species (ROS) in the kidney.

**Figure 9 biomedicines-11-00093-f009:**
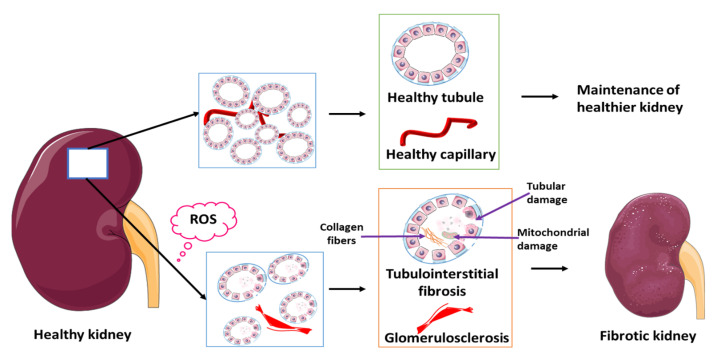
ROS associated with renal fibrosis. Renal fibrosis is mainly associated with inflammation however, it is now widely accepted that ROS also contributes significantly to renal fibrosis. Ischemia/reperfusion is associated with the production of ROS. Ischemia/reperfusion is also one of the major factors associated with the induction of RPN. During fibrosis, renal tubulointerstitial fibrosis, renal parenchyma collapses due to the accumulation of collagen fibers and inflammatory markers. The glomerulus can be scarred and collapse leading to proteinuria. Excessive ROS production can lead to mitochondrial dysfunction which exacerbates injury of kidney tubules. ROS-reactive oxygen species.

**Figure 10 biomedicines-11-00093-f010:**
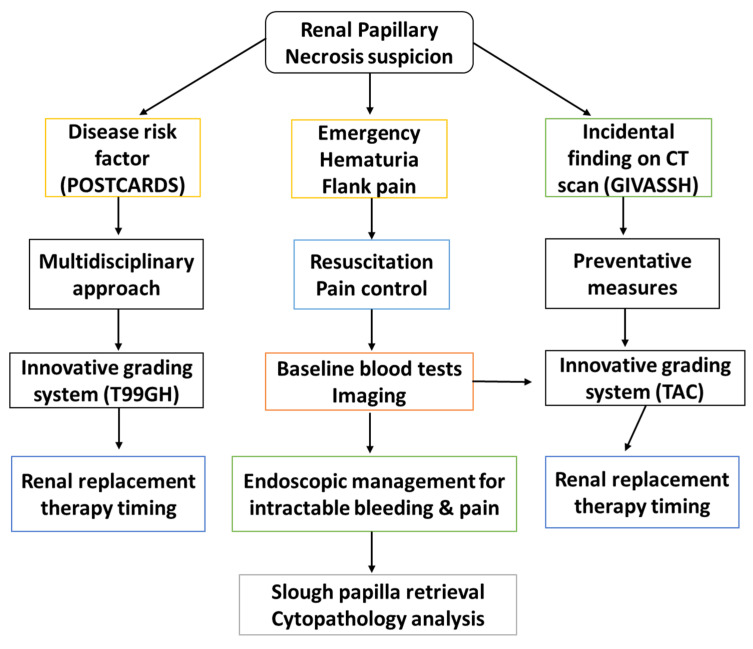
Summary figure depicting RPN proposed algorithm management. Besides identifying risk factors and the role of oxidative stress associated with RPN, it is essential that a multidisciplinary approach is considered for early detection of the disease. Integrated management will ensure successful disease outcomes. TAC: total antioxidant capacity; T99GH: technetium 99 glucoheptonate.

**Table 1 biomedicines-11-00093-t001:** The mnemonic for renal papillary necrosis (RPN) is “POSTCARDS”.

P	Pyelonephritis
O	Obstruction of the urinary tract (e.g., Benign prostatic hyperplasia, [47])
S	Sickle cell hemoglobinopathies, including sickle cell trait
T	Tuberculosis
C	Cirrhosis of the liver, chronic alcoholism
A	Analgesic abuse
R	Renal transplant rejection, radiation
D	Diabetes mellitus
S	Systemic vasculitis

**Table 2 biomedicines-11-00093-t002:** Identified factors associated with renal papillary necrosis (RPN) in Africa.

G	Genetic disorders, other than sickle cell hemoglobinopathies
I	Iatrogenic
V	Viral
A	Antimicrobials
S	Schistosomiasis
S	Substance abuse
H	Hypertension

## Data Availability

Not applicable.

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
