# Peer review of "Renal Papillary Necrosis (RPN) in an African Population: Disease Patterns, Relevant Pathways, and Management"

_biomedicines, 2022, doi:10.3390/biomedicines11010093_

Round 1

Reviewer 1 Report (Previous Reviewer 1)

Re-Review of the manuscript biomedicines-2058075 (previously Biomedicines-1945992)

Renal papillary necrosis (RPN) in an African population: disease patterns, relevant pathways and management

By  Guy Roger Gaudji et al.

Thank you for submitting the next, revised version of the manuscript.

The authors mostly addressed my comments and given in the first review.

I believe that the current version of the manuscript is almost ready for publication.

However, I still have some comments and doubts:

1. The authors introduced a short fragment on the pathogenesis of RPN in the course of post-analgesic nephropathy in the current version of the manuscript - lines 487-502. Paracetamol, aspirin and phenacetin were mentioned. But what about other NSAIDs? There is a question if diclofenac, ketoprofen, acemetacin, and other NSAIDs are not used in the African population? Are there only two studies of NSAID analgesic nephropathy in the African population mentioned in lines 503-513?

2. Moreover, in the abovementioned section (Pathogenesis of RPN in the course of analgesic nephropathy;  lines 487-513), the pathogenesis of post-analgesic nephropathy resulting from drug-induced disorders of renal hemodynamics should be described to a greater extent.

3. In the Summary and Key points section, those factors that are particularly relevant to the African population could be more clearly identified and listed explicitly. Although there is a reference to Table 2, a brief description of the "ideal" patient at risk of developing RPN due to the presence of specific risk factors specific to the African population would be interesting.

Author Response

Dear Reviewer,

Thank you for taking the time to read and make recommendations for the manuscript.

See attached

Reviewer 2 Report (Previous Reviewer 2)

I would like to thank the editorial board for an opportunity to review the revised manuscript. I really appreciate the effort of the authors to try to address my comments and recommendations. However, the revised version still has several weaknesses. Although, the author is focusing on characteristics of RPN particularly in African population, most of details in this manuscript are generally vague and mostly related to “ATN”, “kidney inflammation and fibrosis” but not specifically to RPN nor RPN in African population. The authors tried to emphasize “GIVASH” as a mnemonic to help sort the etiology of RPN in African but the message is not convincing. Most of the time, the information in each compartment of GIVASH are broad and not connected to the main theme. I do suggest that the author may have to focus on “RPN” and less on other issue in every component of GIVASH. For example, the author add a lot of information about nephrotoxicity and HAART which we all know but how is it related to RPN in African? The message needs to be more “concise” and “clear”. However, I do appreciate that the author are trying to revise “diagnosis” part which I think is rather important. However, for figure 5, the author should revamp it and use as a “diagnostic paradigm” for RPN. As of now, this is so confusing. What do you mean by traditional methods? And when would you use endo urological procedure. This is what clinician would want to know. However, this figure does not deliver this message clearly. Lastly, the management scheme on Figure 10 is also unclear. I though the author endorsed GIVASSH over POSTCARDs and now both of them are incorporated into the treatment regimen. Basically, this figure should be a summary of RPN management and Table 10 should be cited in text in which it is not yet.

Author Response

Dear Reviewer,

Thank you for taking the time to read and make recommendations for the manuscript.

See attached

Reviewer 3 Report (New Reviewer)

The authors describe the pathogenesis of renal papillary necrosis (RPN) and mention the shortcomings of the diagnostic criteria. They focused on explaining the risk factors of RPN, particularly in African population, and furthermore, they propose a management algorithm due to the disadvantages of invasive diagnostic methods. Also, the authors adequately addressed the comments of the reviewers, however, there are some minor points that need to be corrected in the text and mentioned below.  

Minor points

1.    Figure 1A. The word “medulla” must be written in lower case. 

2.    Line 144. Say “50year” should be “50 year”

3.    Line 147. The abbreviation for HIV should be defined the first time it is mentioned. 

4.    Line 609. Say “practise” should say “practice”

5.    Figure 5,8 and 10. The letters in figures 5,8 and 10 are blurred, please improve the image quality. 

6.    Line 514. In section 8. Reactive oxygen species and RPN, add endoplasmic reticulum stress as an endogenous source of ROS and briefly explain in the text and include them in figure 8. 

7.    Line 627. The abbreviation of Keap1 must be written in lower case. 

8.    Figure 9. Say “heathier” should say “healthier”

9.    Line 722. The word “management” must be written in lower case. 

10. Line 727. Say “alkalinisation” should say “alkalinization”

11. Line 731. The reference 19 should not be written in superscript.

12. The authors could combine sections 11 and 12 and make a section of conclusions and future directions with those ideas. 

Author Response

Dear Reviewer,

Thank you for taking the time to read and make recommendations for the manuscript.

See attached

This manuscript is a resubmission of an earlier submission. The following is a list of the peer review reports and author responses from that submission.

Round 1

Reviewer 1 Report

Review of the manuscript Biomedicines-1945992

Renal papillary necrosis (RPN) in an African population: disease patterns, relevant pathways and management

By  Guy Roger Gaudji et al.

The manuscript is a narrative review and describes the epidemiology, diagnostic criteria, and risk factors for the development of renal papillary necrosis (RPN). In the Introduction, Authors precede the main substantive part with a short description of the anatomical and physiological aspects related to the renal papillae, illustrating these issues with gross anatomy morphology photos and a figure showing renal vascularization. The manuscript also includes macroscopic pictures of kidneys with RPN and histopathological images. In the further part of the article, Authors briefly discuss the role of oxidative stress in the pathogenesis of RPN. At the end of the manuscript, the proposed algorithm of diagnostic and therapeutic management in the RPN is characterized. The paper  includes 106 references, mostly from the last 10 years.

In my opinion, the article is interesting, gives the impression of being well thought out  and organized and has a good graphic design. An interesting aspect is the proposed acronym GIVASSH, which is to express specific etiological factors of RPN in patients from Africa / South Africa and complement the classic etiological factors described by the acronym POSTCARDS.

Nevertheless, reading the article raises some comments:

1. Throughout the manuscript, in several places (eg, lines 41, 172-172 and others), Authors mention drug abuse and - as I understand it - the development of analgesic nephropathy as a risk factor for RPN. Unfortunately, there is no broader fragment in the entire manuscript focusing on the pathogenesis of RPN in the course of analgesic nephropathy. I believe that the manuscript would gain in complexity after introducing a separate chapter describing this issue, with particular emphasis on data on the consumption of nephrotoxic analgesics by patients from Africa / South Africa.

2. Introduction line 47 – an information about biomarkers is given. Currently, in laboratory diagnostics in nephrology, protein markers such as KIM-1, NGAL-1, osteopontin, FABP and others are increasingly used. I think it would be worth referring to whether the analysis of these markers is also applicable in the RPN – especially because some of them are used for the diagnosis of AKI/CKD.

3. I do not fully understand why certain risk factors such as I-iatrogenic or A-antimicrobials under the acronym GIVASSH are regarded to be “exclusively” specific to the African population. After all, aminoglycosides, quinolones and carbapenems are used globally, especially in Europe. Iatrogenic damage happens all over the world. Hence, I believe that this fragment should need a more detailed explanation.

4. Chapter 8 - Reactive oxygen species and RPN is very sketchy and it would need to be extended and redrafted. The production of reactive oxygen species and reactive nitrogen species in the kidneys should be described in more details, as well as the altered “cytokine storm” contributing to the maintenance of low-grade inflammation and, consequently, to the development of kidney fibrosis. Moreover, in view of the partially disclosed mechanisms of oxidative stress in the kidney, are there potential pharmacological sites of action to reduce kidney damage? Authors should briefly refer to oxidative stress mechanisms as potential therapeutic targets and mention OS-alleviating agents acting via their interference with selected pathophysiological pathways, eg. xanthine oxidase inhibitors, nicotinamide adenine dinucleotide phosphate oxidase inhibitors, protein kinase C inhibitors, transforming growth factor β inhibitors, or activators of nuclear factor erythroid 2-related factor 2 etc.

Reviewer 2 Report

I would like to thank the editor for an opportunity to review this article title “Renal papillary necrosis (RPN) in an African population: disease patterns, relevant pathways and management”. Overall the author try to provide a comprehensive review of RPN particularly in African population. While RPN is mainly a result of severe ischemic injury in the renal papillae, the author should provide “specific” information pertaining to RPN rather than “general” ischemic insults. Particularly in #8. ROS and RPN, in which, the general theme of this section is non-specific. The author should also provide more information regarding treatment and outcome in patients with RPN. For example, how many patients require dialysis, rate of CKD after RPN and transplant etc.

Minor point

1.      Line: 82-> vas recta; please correct

2.      Please clarify/re-write the sentences between line 113-118. Does the author mean that some patients may be asymptomatic especially in the kidney transplant patient who have acute rejection?

3.      Line 136; “A condition known to exacerbate kidney disease.” This is incomplete sentence. Does the author want to add some information about certain factors?

4.      Line 211; please change from recessive to dominant

5.      Line 214; PKHD1 is associated with ARPKD  nor ADPKD

6.      Spell out all abbreviations at their first use

7.      Line 247; please provide supporting evidence of HIV and infections and RPN. Now, it is rather a general information about HIV in Africa.

8.      Line 257; “Previous studies on the 257 accuracy of the Cockcroft Gault equation in HIV patients yielded conflicting results.” Please elaborate? Is it relevant to this section?

9.      Line 283 ; Again, please show evidence and provide study of schistosomiasis associated RPN

10.   Please provide abbreviation list in each figures and tables

11.   In RPN management-> please explain why alkalizination of urine is beneficial in RPN?

12.   Can author provide an example of imaging findings in patients with RPN

13.   Can author elaborate more why RPN is more prevalent in Australia and England compared to the US. In my understanding, RPN is not a common disease and to say that 15-20% of patients undergoing kidney transplant have RPN seems a bit high.

14.   Figure 2D-> please use number with arrow to demonstrated ghost cells and foamy macrophages. Please also provide more characteristics descriptions of these cells in the Figure legend.

15.   How dose pelvic exam is helping in diagnosis of RPN?

16.   Figure 3: There are many tests the authors listed here in the RPN diagnosis but most of them are not POCT test for example cystoscopy, pyelogram, cytology and etc. Please be more specific since the title of the figure is talking about POCT